## Comment

ecology, evolution, theoretical biology

**Author for correspondence:**
David Bahry
e-mail: davidbahry@cmail.carleton.ca

# Senescence, trait parameterization and (st)age-specific forces of selection

David Bahry

Department of Biology, Carleton University, 1125 Colonel By Dr, Ottawa, ON, Canada

DB, 0000-0002-3342-9597

Roper *et al.* [1] provide a valuable review of recent work in the evolutionary theory of senescence (ageing), going beyond Hamilton's age-specific indicators of selection [2] to consider when and why fitness sensitivities might not always decrease with age, including the importance of more generally stage-structured life histories [3–5]. However, they repeat a subtle mischaracterization of the relationship between Hamilton's indicators of selection and Caswell's generalized fitness sensitivity for (st)age-structured population projection matrices [6,7]. This comment intuitively explains the source of the discrepancy and gives the true relationship. It is connected to Baudisch's [8,9] insight that different trait parameterizations require different fitness sensitivities: Hamilton implicitly assumed that genes multiply survival probabilities; Caswell implicitly assumed that genes add to and subtract from survival probabilities.

Senescence evolves because natural selection 'cares less about' late life than about early life [10,11]. This idea was developed heuristically by Haldane [12], Medawar [13] and Williams [14], then given formal age-structured demographic justification by Hamilton [2]. When and how Hamilton's framework is appropriate for Mendelian population genetics has been investigated by Charlesworth [15,16].

An age-structured life history [11] is a schedule of age $x$; of a newborn's probability of still being alive at that age, $l_x$; and of her instantaneous rate of production of offspring at that age given she is alive, $b_x$; for simplicity assume asexually reproducing female populations. Life histories can also be modelled for discrete intervals, where the simplest case is a birth-pulse population with all reproduction occurring on the birthday; this assumption makes the following equations exact [2, p. 16]; this comment assumes a pre-birth census [5, p. 403]. Let an individual be aged $x$ entering interval $X$ of unit length; $l_X = l_x$ be a newborn's probability of survival to the beginning of interval $X$; and $m_X$ be the number of daughters she produces at the beginning of interval $X$, given she is alive.

Further quantities can be derived: interval age-specific survival probability is $p_X = l_{X+1}/l_X$; interval age-specific probability of death is $q_X = [l_X - l_{X+1}]/l_X$; instantaneous force of mortality, or hazard rate, is $\mu_x = -l'_x/l_x$; and effective fecundity, the expected number of daughters to a mother aged $x$ at time $t$ surviving to be censused for the first time at time $t + 1$, for pre-birth censusing is $F_X = m_X p_0$. Lifetime reproductive output, noting that $m_0 = 0$, is

$$R_0 = \sum_{X=0}^{\infty} l_X m_X, \tag{1.1}$$

the fraction entering each age class, once the population reaches stable age distribution, is

$$c_X = \frac{e^{-rX} l_X}{\sum_{Y=0}^{\infty} e^{-rY} l_Y}, \tag{1.2}$$

and age-specific reproductive value, an individual's expected further contribution to the ancestry of far-future generations relative to the expected contribution of a newborn, is

$$V_X = \frac{v_X}{v_0} = \sum_{Y=X}^{\infty} e^{-r(Y-X)} \frac{l_Y}{l_X} m_Y, \tag{1.3}$$

where $r$ is the intrinsic rate of increase, the population's exponential growth rate once in stable age distribution, calculated as the unique real root of the Euler–Lotka equation:

$$1 = \sum_{X=0}^{\infty} e^{-rX} l_X m_X. \qquad (1.4)$$

Analogous quantities can be defined using $F_X$ instead of $m_X$, but then the resulting time lag from censusing parents to counting offspring complicates notation [17, p. 814]. A stable population grows each time step by $\lambda = e^r$.

Hamilton [2] considered hypothetical genes that, for one age interval $X$, add a bit to interval fecundity $m_X$, or multiply by a bit interval survival probability $p_X$, i.e. add to its logarithm $\ln p_X$, i.e. subtract from the average instantaneous hazard rate over the interval, $\bar{\mu}_X$ (since $p_X = e^{-\bar{\mu}_X}$). He then asked how much such genes would affect fitness, and how this depends on the age interval $X$ at which the gene's effect occurs. This embodies the general approximation

$$\Delta(\text{fitness}) \approx \Delta(\text{trait}) \times \frac{\partial(\text{fitness})}{\partial(\text{trait})}. \qquad (1.5)$$

The partial derivative on the right is a fitness sensitivity, or selection gradient, or indicator of the force of selection. Applying this approximation requires choosing the fitness measure ($r$ versus $\lambda$ versus $R_0$ versus …), the scalar trait (mortality versus wingspan versus …), and how to parameterize the trait (log wingspan versus square root of wingspan versus …). Hamilton assumed fitness is $r$ and, as mentioned, considered age-specific interval survival and fecundity, parameterized as $\ln p_X$ and $m_X$; he also analysed the analogous continuous-time case, but noted that the fitness effect of an infinitely brief change to instantaneous demographic rates is zero. This determined which sensitivities he sought; he found formulae for them by implicitly differentiating the Euler–Lotka equation:

$$-\frac{\partial r}{\partial \bar{\mu}_X} = \frac{\partial r}{\partial \ln p_X} = \frac{\sum_{Y=X+1}^{\infty} e^{-rY} l_Y m_Y}{T} \qquad (1.6a)$$

and

$$\frac{\partial r}{\partial m_X} = \frac{e^{-rX} l_X}{T}, \qquad (1.6b)$$

where the denominator $T = \sum_{Y=0}^{\infty} Y e^{-rY} l_Y m_Y$ is generation length, defined as the mean age of mothers; it is not a function of $X$ and can be ignored for our purposes. The numerators are monotonically decreasing functions of the interval $X$ at which the gene acts, vindicating the insight of Haldane, Medawar and Williams that natural selection cares less about late than about early life (with the minor exception that the second, equation (1.6b), can rise with age in rapidly shrinking populations; these tend to go extinct anyway [15, p. 192]). Different assumptions imply different fitness sensitivities, however. For instance, in stationary populations, maintained at constant size by certain forms of density dependence, fitness can be measured as $R_0$ [18]; thus Dańko *et al.* used the sensitivity $(-\partial R_0 / \partial \bar{\mu}_X)$ [19]. Baudisch, assuming fitness as $r$, but genes that multiplicatively reduce interval average hazard rate, thus used the sensitivity $(-\partial r / \partial \ln \bar{\mu}_X)$ [8,9].

Hamilton's assumption that genes multiply $p_X$, i.e. subtract from $\bar{\mu}_X$ (additive hazards), was intuitive [15, p. 191]: it follows if genes add and remove probabilistically

independent mortality risks. However, despite intuition, Baudisch [9, p. 24–26] cites evidence that genes multiplying $\bar{\mu}_X$ (proportional hazards) may also be common. Interestingly, Baudisch's proportional hazards indicator relates to Hamilton's additive hazards indicator as

$$-\frac{\partial r}{\partial \ln \bar{\mu}_X} = \bar{\mu}_X \left( -\frac{\partial r}{\partial \bar{\mu}_X} \right). \qquad (1.7)$$

This, unlike equation (1.6a), can in fact rise with age under some conditions; however, it still must decrease with age in initially non-senescing (constant $\bar{\mu}_X$) or negatively senescing (decreasing $\bar{\mu}_X$) populations and therefore seems intuitively compatible with the ubiquity of senescence in nature. The economist and demographer Ronald D. Lee has also modified Hamilton's framework to take into account parental and grandparental resource transfers, of relevance to the evolution of post-reproductive lifespan, including in humans and in whales [20,21].

From his results, Hamilton [2, pp. 12 and 35] suggested the possibility 'that senescence is an inevitable outcome of evolution', though then acknowledging that 'To what extent and in exactly what way life schedules will be moulded my natural selection depends on what sort of genetical variation is available'. However, senescence is not universal [22,23]; freshwater hydras are the most well-known exception [24]. It is therefore a task for theoretical and empirical, evolutionary and proximate, biogerontology to account for the diversity of both senescent and non-senescent life histories across the tree of life [25,26]. Roper *et al.* [1] is a valuable review of some recent work in this area, including on the importance of stage structure [3–5].

In stage-structured life histories, vital rates depend on life cycle stage, which may or may not correspond to and is more general than chronological age. For instance, fecundity in a jellyfish might depend on whether it is a polyp or a medusa, and the jellyfish might be able to cycle between these stages, whereas a 21-year-old cannot return to being 20 years old [27]. Importantly, a stage-structured life history can spend different amounts of time in different stages; for instance, fitness in loggerhead sea turtles can be more sensitive to mortality in large juveniles than in hatchlings, with implications for conservation strategy, simply because they spend longer as large juveniles than as hatchlings [28]. Note that if we model evolution, not only ecology, as stage-structured, we are implicitly assuming genes tend to affect vital rates over entire stages rather than at single ages: for instance, that a gene reduces mortality for all approximately 7 years spent as a large juvenile, not just for one of them. Stage structure may be especially important for plant evolution [29].

Stage-structured life histories are modelled in discrete time using population projection matrices [3–5]; discrete time age-structured life histories are a special case, with the age-structured projection matrix known as the Leslie matrix. The population projection matrix **A** is a grid of numbers, whose entries $a_{ij}$ represent the contribution an individual in stage $j$ at time $t$ makes to stage $i$ at time $t+1$, whether by surviving and remaining in the stage; surviving and transitioning to the stage; or by giving birth to surviving offspring. For the age-structured special case, the Leslie matrix's top row contains the effective fecundities, starting from $F_1 = a_{1,1}$; its sub-diagonal contains the survival probabilities, starting

from $p_1 = a_{2,1}$; and all other entries are zero. Right-multiplying a vector of the number alive now in each (st)age, $\mathbf{n}$, gives the abundances for the next census: $\mathbf{n}(t+1) = \mathbf{An}(t)$; with a pre-birth census newborns aged $x = 0$ are not censused in $\mathbf{n}$ at $t$, but are reckoned part of their mothers' fecundity. The stable (st)age distribution is a right eigenvector $\mathbf{w}$ of the matrix $\mathbf{A}$; the (st)age-specific individual reproductive values also form a left eigenvector $\mathbf{v}$ of the matrix $\mathbf{A}$; the dominant eigenvalue is $\lambda$, the stable population's growth per time step. Caswell [6] showed that the sensitivity of fitness $\lambda$ to small additive changes to any entry $a_{ij}$ can be calculated from the eigenvectors; in the notation of [3],

$$\frac{\partial \lambda}{\partial a_{ij}} = \frac{w_j v_i}{\langle \mathbf{w}, \mathbf{v} \rangle}, \tag{1.8}$$

where $w_j$ is relative abundance in the stable population of (st)age $j$; $v_i$ is relative reproductive value of individuals in (st)age $i$; and $\langle \mathbf{w}, \mathbf{v} \rangle$ is the eigenvectors' dot product. Caswell summarizes: 'The effect (on $\lambda$) of a change in $a_{ij}$ is proportional to the reproductive value of the destination stage and to the abundance of the origin stage in the stable population' [7, p. 529]. Applied to the age-structured Leslie matrix, equation (1.8) becomes proportional to

$$\frac{\partial \lambda}{\partial p_X} \propto c_X V_{X+1} \tag{1.9a}$$

and

$$\frac{\partial \lambda}{\partial F_X} \propto c_X V_1. \tag{1.9b}$$

Although these are related to Hamilton's indicators, they are not identical: purely conventionally, they use $\lambda$ rather than $r$ for fitness, and effective fecundity $F_X$ rather than fecundity $m_X$; more substantively, equation (1.9a) assumes additive genetic effects on survival probability, while equation (1.6a) assumes multiplicative genetic effects on survival probability; see [2, p. 17]. Thus, Roper et al. repeat a subtle mischaracterization by Caswell [7, p. 534], when they describe equations (1.9a) and (1.9b) as a 'reformulation of' equations (1.6a) and (1.6b),

and write, still implying equivalence, that 'For any given (st)age of a life cycle, the force of selection on an increase in the reproduction or mortality of that (st)age is proportional to the product of two key components: (i) the stable age distribution of individuals at that (st)age and (ii) the reproductive value of individuals that the (st)age contributes to the population (new offspring or their surviving selves)' [1, p. 5]. The true relationships of Hamilton's indicators to Caswell's fitness sensitivity applied to the Leslie matrix, assuming a pre-birth census, are (electronic supplementary material, 'Mathematical appendix'):

$$\frac{\partial r}{\partial \ln p_X} = \frac{p_X}{\lambda}\left(\frac{\partial \lambda}{\partial p_X}\right) = \frac{w_X RRV_X}{T} \propto c_X RRV_X \tag{1.10a}$$

and

$$\frac{\partial r}{\partial m_X} = \frac{p_0}{\lambda}\left(\frac{\partial \lambda}{\partial F_X}\right) = \frac{w_X V_0}{T} \propto c_X V_0, \tag{1.10b}$$

where $RRV_X = V_X - m_X$ is residual reproductive value; reproductive value at birth is $V_0 = 1$; and $w_X = e^{-rX} l_X \propto c_X$; note that $RRV_X \neq V_{X+1}$ [11]. Caswell's insight, modified to account for the difference between birth and first census, remains that equation (1.10b) 'shows why reproductive value is apparently missing from' equation (1.6b): 'reproductive value at birth … is scaled to equal 1' [7, p. 540]. For short intervals with low per-interval fecundity, $p_X \approx 1$ and $V_X \approx V_{X+1} \approx RRV_X$; therefore despite the imprecision of equating Caswell's sensitivity with Hamilton's indicators, quantitatively, the difference is minor; this may account for the qualitative agreement with their proportionality claim that Roper et al. observe in life-history data [1, pp. 4–5], even if Hamilton's genetic assumptions are the correct ones. If Baudisch's genetic assumptions are the correct ones [8,9], her sensitivity of fitness $r$ to proportional hazards (equation 1.7) equals $\bar{\mu}_X w_X RRV_X / T \propto \bar{\mu}_X c_X RRV_X$.

Data accessibility. This article has no additional data.

Competing interests. I declare I have no competing interests.

Funding. I received no funding for this study.

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
