## [Peer Review File · Proceedings of the Royal Society B: Biological Sciences]

Review History

RSPB-2021-1910.R0 (Original submission)

Review form: Reviewer 1

Recommendation

Accept with minor revision (please list in comments)

Scientific importance: Is the manuscript an original and important contribution to its field?

Good

General interest: Is the paper of sufficient general interest?

Good

Quality of the paper: Is the overall quality of the paper suitable?

Excellent

Is the length of the paper justified?

Yes

Should the paper be seen by a specialist statistical reviewer?

No

Do you have any concerns about statistical analyses in this paper? If so, please specify them explicitly in your report.

No

It is a condition of publication that authors make their supporting data, code and materials available - either as supplementary material or hosted in an external repository. Please rate, if applicable, the supporting data on the following criteria.

Is it accessible?

N/A

Is it clear?

N/A

Is it adequate?

N/A

Do you have any ethical concerns with this paper?

No

Comments to the Author

As a comment of Roper et al.'s paper, Bahry provides a neat explanation of the relationship between Hamilton's indicators of selection and Caswell's generalized fitness sensitivity. The equations 3.1 and 3.2 in Roper et al.'s paper rely on the false assumption that Hamilton and Caswell use the same fitness measure, the same fecundity measure, and the same genetic effect on survival probabilities. Bahry provide the true expression and the approximations of these equations 3.1 and 3.2 (equations 10.1 and 10.2 in Bahry's paper). Bahry shows however that despite these false assumptions, the difference is minor. As a result, the claim of Roper et al that senescence pattern could be inferred from population structure and reproduction value still holds qualitatively.

In my review of Roper et al. paper, I acknowledged that I was just discovering Caswell's derivations. I am happy to read these additional explanations about it; and I feel a bit ashamed that I took for granted the same false assumptions than Roper et al. made without thinking about it further.

This comment is very well written, and I haven't much to recommend except perhaps that the author could phrase to what extent Roper et al.'s conclusion holds if we consider the sensitivity of fitness to proportional hazards, instead of finishing abruptly with a mathematical expression (line 157).

I am still a bit confused about the discrepancies between Hamilton's conclusion ("senescence is universal" on the basis of the numerators decreasing monotonically in the expressions because the probability to survive to a specific age decreases at this specific age increases) with the one by Roper et al (senescence depends on the population age structure and reproductive value). Bahry argue that "In stage-structured life histories, vital rates depend on life cycle stage, which may or may not correspond to and is more general than chronological age" (lines 95-96). Nonetheless in the data shown by Roper et al, it seems that those stages are somehow correlated with chronological age, so even if the time spent in each stage may differ, I don't fully understand why Hamilton's conclusion does not hold when interpreting equations 10.1 and 10.2 in Bahry's paper. If the editor or Dr. Bahry think that other readers might still be confused as I am, perhaps Dr. Bahry could take the opportunity to be crystal-clear on that point in this commentary.

Review form: Reviewer 2

Recommendation

Accept as is

Scientific importance: Is the manuscript an original and important contribution to its field?

Excellent

General interest: Is the paper of sufficient general interest?

Excellent

Quality of the paper: Is the overall quality of the paper suitable?

Excellent

Is the length of the paper justified?

Yes

Should the paper be seen by a specialist statistical reviewer?

No

Do you have any concerns about statistical analyses in this paper? If so, please specify them explicitly in your report.

No

It is a condition of publication that authors make their supporting data, code and materials available - either as supplementary material or hosted in an external repository. Please rate, if applicable, the supporting data on the following criteria.

Is it accessible?

N/A

Is it clear?

N/A

Is it adequate?

N/A

Do you have any ethical concerns with this paper?

No

Comments to the Author

The mathematical theories proposed to understand are quite complicated, and even experts often talk over each other and make mistakes in presentation. This Comment is a really wonderful contribution pointing out a misinterpretation in Roper et al. that is important to stress if we are to move forward in this field. Caswell's contributions to the field of senescence are important, but poorly understood and the author makes a good case for his interpretation. This should be very valuable.

Decision letter (RSPB-2021-1910.R0)

13-Oct-2021

Dear Mr Bahry

I am pleased to inform you that your Review manuscript RSPB-2021-1910 entitled "Senescence, trait parameterization, and (st)age-specific forces of selection" has been accepted for publication in Proceedings B.

The referee(s) do not recommend any further changes. Therefore, please proof-read your manuscript carefully and upload your final files for publication. Because the schedule for publication is very tight, it is a condition of publication that you submit the revised version of your manuscript within 7 days. If you do not think you will be able to meet this date please let me know immediately.

To upload your manuscript, log into <http://mc.manuscriptcentral.com/prsb> and enter your Author Centre, where you will find your manuscript title listed under "Manuscripts with Decisions." Under "Actions," click on "Create a Revision." Your manuscript number has been appended to denote a revision.

You will be unable to make your revisions on the originally submitted version of the manuscript. Instead, upload a new version through your Author Centre.

1) A text file of the manuscript (doc, txt, rtf or tex), including the references, tables (including captions) and figure captions. Please remove any tracked changes from the text before submission. PDF files are not an accepted format for the "Main Document".

2) A separate electronic file of each figure (tiff, EPS or print-quality PDF preferred). The format should be produced directly from original creation package, or original software format. Please note that PowerPoint files are not accepted.

3) Electronic supplementary material: this should be contained in a separate file from the main text and the file name should contain the author's name and journal name, e.g. `authorname_procb_ESM_figures.pdf`

All supplementary materials accompanying an accepted article will be treated as in their final form. They will be published alongside the paper on the journal website and posted on the online figshare repository. Files on figshare will be made available approximately one week before the accompanying article so that the supplementary material can be attributed a unique DOI. Please see: <https://royalsociety.org/journals/authors/author-guidelines/>

4) Data-Sharing and data citation

It is a condition of publication that data supporting your paper are made available. Data should be made available either in the electronic supplementary material or through an appropriate repository. Details of how to access data should be included in your paper. Please see <https://royalsociety.org/journals/ethics-policies/data-sharing-mining/> for more details.

<http://datadryad.org/submit?journalID=RSPB&manu=RSPB-2021-1910> which will take you to your unique entry in the Dryad repository.

Once again, thank you for submitting your manuscript to Proceedings B and I look forward to receiving your final version. If you have any questions at all, please do not hesitate to get in touch.

Sincerely,
Dr The Proceedings B Team
mailto:proceedingsb@royalsociety.org

Associate Editor Board Member: 1

Comments to Author:

Both referees agree that the points are valid and make a positive contribution to the discussion.

Reviewer(s)' Comments to Author:

Referee: 1

Comments to the Author(s)

As a comment of Roper et al.'s paper, Bahry provides a neat explanation of the relationship between Hamilton's indicators of selection and Caswell's generalized fitness sensitivity. The equations 3.1 and 3.2 in Roper et al.'s paper rely on the false assumption that Hamilton and Caswell use the same fitness measure, the same fecundity measure, and the same genetic effect on survival probabilities. Bahry provide the true expression and the approximations of these equations 3.1 and 3.2 (equations 10.1 and 10.2 in Bahry's paper). Bahry shows however that despite these false assumptions, the difference is minor. As a result, the claim of Roper et al that senescence pattern could be inferred from population structure and reproduction value still holds qualitatively.

In my review of Roper et al. paper, I acknowledged that I was just discovering Caswell's derivations. I am happy to read these additional explanations about it; and I feel a bit ashamed that I took for granted the same false assumptions than Roper et al. made without thinking about it further.

This comment is very well written, and I haven't much to recommend except perhaps that the author could phrase to what extent Roper et al.'s conclusion holds if we consider the sensitivity of fitness to proportional hazards, instead of finishing abruptly with a mathematical expression (line 157).

I am still a bit confused about the discrepancies between Hamilton's conclusion ("senescence is universal" on the basis of the numerators decreasing monotonically in the expressions because the probability to survive to a specific age decreases at this specific age increases) with the one by Roper et al (senescence depends on the population age structure and reproductive value). Bahry argue that "In stage-structured life histories, vital rates depend on life cycle stage, which may or may not correspond to and is more general than chronological age" (lines 95-96). Nonetheless in the data shown by Roper et al, it seems that those stages are somehow correlated with chronological age, so even if the time spent in each stage may differ, I don't fully understand why Hamilton's conclusion does not hold when interpreting equations 10.1 and 10.2 in Bahry's paper. If the editor or Dr. Bahry think that other readers might still be confused as I am, perhaps Dr. Bahry could take the opportunity to be crystal-clear on that point in this commentary.

Referee: 2

Comments to the Author(s)

The mathematical theories proposed to understand are quite complicated, and even experts often talk over each other and make mistakes in presentation. This Comment is a really wonderful contribution pointing out a misinterpretation in Roper et al. that is important to stress if we are to move forward in this field. Caswell's contributions to the field of senescence are important, but poorly understood and the author makes a good case for his interpretation. This should be very valuable.

Decision letter (RSPB-2021-1910.R1)

18-Oct-2021

Dear Mr Bahry

I am pleased to inform you that your manuscript entitled "Senescence, trait parameterization, and (st)age-specific forces of selection" has been accepted for publication in Proceedings B.

Your article has been estimated as being 3 pages long. Our Production Office will be able to confirm the exact length at proof stage.

Data Accessibility section

Open Access

Paper charges

Sincerely,
